# Intrinsic supercurrent non-reciprocity coupled to the crystal structure of a van der Waals Josephson barrier

Jae-Keun Kim [1,4] ✉, Kun-Rok Jeon [2,4], Pranava K. Sivakumar[1], Jaechun Jeon[1], Chris Koerner[3], Georg Woltersdorf [3] & Stuart S. P. Parkin [1] ✉

Non-reciprocal electronic transport in a spatially homogeneous system arises from the simultaneous breaking of inversion and time-reversal symmetries. Superconducting and Josephson diodes, a key ingredient for future non-dissipative quantum devices, have recently been realized. Only a few examples of a vertical superconducting diode effect have been reported and its mechanism, especially whether intrinsic or extrinsic, remains elusive. Here we demonstrate a substantial supercurrent non-reciprocity in a van der Waals vertical Josephson junction formed with a $T_d$-WTe$_2$ barrier and NbSe$_2$ electrodes that clearly reflects the intrinsic crystal structure of $T_d$-WTe$_2$. The Josephson diode efficiency increases with the $T_d$-WTe$_2$ thickness up to critical thickness, and all junctions, irrespective of the barrier thickness, reveal magneto-chiral characteristics with respect to a mirror plane of $T_d$-WTe$_2$. Our results, together with the twist-angle-tuned magneto-chirality of a $T_d$-WTe$_2$ double-barrier junction, show that two-dimensional materials promise vertical Josephson diodes with high efficiency and tunability.

The non-reciprocal behavior in dissipative current flows, known as the diode effect, has played a central role in modern electronic devices and circuits[1,2]. In conventional schemes, non-reciprocity along the current direction arises from spatial inhomogeneity[1,2]. Recently, it has been shown that when inversion and time-reversal symmetries are both broken, and in combination with a spin-orbit interaction (SOI), even spatially homogeneous systems can provide for diode functionality[3–5]. By implementing this magneto-chirality with superconductors (SCs) and matching the superconducting gap with the SOI energy[6,7], one can achieve a directional, non-dissipative, supercurrent flow, which is a prerequisite for the realization of future superconducting quantum devices.

To date, several methods at the materials and device levels have been developed to intrinsically and/or extrinsically break the inversion symmetry. For example, via artificially patterned conformal arrays of nanoscale holes in a MoGe film[8], via ionic-liquid gating of two-dimensional (2D) MoS$_2$ flakes, and via the formation of non-centrosymmetric V/Nb/Ta superlattices or using a few layers of 2H-NbSe$_2$ have enabled the observation of non-reciprocal critical currents near their corresponding superconducting transition temperatures $T_c$[9–11]. Notably, non-reciprocity in Josephson supercurrents have also been realized in lateral Josephson junctions (JJs) with structures: Al/InAs 2D electron gas (2DEG)/Al[7], Nb/proximity-magnetized Pt/Nb[12], Nb/NiTe$_2$ type-II Dirac semimetal/Nb[13], three-terminal Al/InAs 2DEG/Al JJs[14]. To break the inversion symmetry, the first two utilize a Rashba(-type) superlattice and barrier, respectively, the third one exploits topological surface states in a NiTe$_2$ barrier, whereas the last one employs the geometric breaking of the device's inversion symmetry by the presence of third terminal. On the other hand, there have been only a few relevant studies on vertical JJs. A field-free Josephson diode effect has been recently observed in vertical NbSe$_2$/Nb$_3$Br$_8$/NbSe$_2$ junctions[15], but the non-reciprocal supercurrents appear in the

[1]Max Planck Institute of Microstructure Physics, Weinberg 2, 06120 Halle (Saale), Germany. [2]Department of Physics, Chung-Ang University (CAU), Seoul 06974, Republic of Korea. [3]Department of Physics, Martin Luther University Halle-Wittenberg, Von-Danckelmann-Platz 3, 06120 Halle, Germany. [4]These authors contributed equally: Jae-Keun Kim, Kun-Rok Jeon. ✉e-mail: jkkim@mpi-halle.mpg.de; stuart.parkin@mpi-halle.mpg.de

direction along which the inversion symmetry is broken and the diode polarity is, moreover, independent of an applied magnetic field, making investigations of other devices highly desirable to unravel the role of van der Waals barriers. Moreover, whether or not there can be an intrinsic Josephson diode effect in such vertical junctions is the outstanding quest in this increasing research field of supercurrent non-reciprocity.

Very recent studies[16,17] on the Al/InAs 2DEG/Al lateral junctions have investigated a possible correlation between the bulk inversion symmetry breaking (i.e. Dresselhaus-type) and the supercurrent non-reciprocity. In the case of interface inversion symmetry breaking (i.e. Rashba-type), the associated spin-orbit (SO) field in $k$-space is fundamentally isotropic[16]. When the Josephson barrier possesses the Dresselhaus-type symmetry breaking in addition to the Rashba-type symmetry breaking, the overall SO field becomes anisotropic, making the magneto-chiral anisotropy of supercurrent non-reciprocity dependent of the Josephson barrier's crystal structure[16]. However, the strength of Dresselhaus-type SO field in such lateral devices turns out to be quite small (<10% at most) as compared with that of Rashba SO field. For this reason, to the best of our knowledge, the lateral device geometry is less suited for the investigation of supercurrent non-reciprocity derived solely from the bulk inversion symmetry breaking of the Josephson barrier.

Here, we utilize $T_d$-WTe$_2$ single-crystal exfoliated flakes as an inherently inversion symmetry breaking barrier in van der Waals (vdW) JJs and through extensive measurements of the $T_d$-WTe$_2$ flake thickness, magnetic field strength/angle and temperature dependences, we demonstrate that the supercurrent non-reciprocity along the vertical direction is intimately connected and is highly dependent on the crystal structure of the $T_d$-WTe$_2$ barrier, thereby allowing a clear distinction between intrinsic and extrinsic mechanisms[7,12,13] and providing the experimental realization of the intrinsic JDE that results *purely* from the bulk inversion symmetry breaking of the $T_d$-WTe$_2$ barrier. Our study establishes that the *co-existence* of the magneto-linearity, the magneto-chirality and the thickness and temperature-scaleable diode efficiency constitute the *intrinsic* supercurrent non-reciprocity, coupled to the crystal structure of a vdW barrier.

## Results

As illustrated in Fig. 1a, we fabricate vdW vertical JJs, in which a $T_d$-WTe$_2$ flake (2–60 nm thick) is sandwiched between two superconducting

2$H$-NbSe$_2$ flakes, in an inert atmosphere glovebox using dry transfer techniques (see Methods for details). Here, we employed the NbSe$_2$ flakes thicker than 10 nm (16–17 × the monolayer thickness) to preclude the unintended contribution of Ising Cooper paring[18] to the non-reciprocal transport properties of WTe$_2$ JJs. Note that $T_d$-WTe$_2$ itself exhibits highly interesting physical properties including a type-II Weyl semi-metallic behavior[19], higher-order topological hinge states[20] and quantum spin-Hall edge states[21]. In the present study, we focus on how the crystal inversion symmetry of $T_d$-WTe$_2$ is reflected in a vertical Josephson diode effect. Due to the lack of a screw rotational symmetry in $T_d$-WTe$_2$ from the Te atoms, the orthorhombic phase $T_d$-WTe$_2$ is non-centrosymmetric[20,22,23]. The mirror-symmetry plane of $T_d$-WTe$_2$ along the $b$-axis (green plane in Fig. 1b) creates a polar axis, that is, an internal crystal electric field $\varepsilon_{cr}$[22] and together with the heavy W atoms provide a strong SOI. This, in conjunction with an external in-plane (IP) magnetic field $\mu_0 H_\parallel$, that breaks the time reversal symmetry, allows for an anomalous phase $\varphi_0$ to be created in the current-phase relationship (CPR) of the vertical JJ[24–26]. As previously discussed theoretically[24–26], the presence of a finite $\varphi_0$ is essential to generate a rectified Josephson supercurrent, namely, unequal positive and negative Josephson switching supercurrents $I_c^+ \neq I_c^-$[7,12].

Below, we will separate a crystal-asymmetry-driven intrinsic diode effect from other extrinsic possibilities in the following manner. When the dc bias current $I$ is applied along the vertical direction (// $c$-axis) and the IP magnetic field is applied in the $a$-$b$ plane of the WTe$_2$ (blue plane in Fig. 1b), the Josephson diode efficiency $\eta = \frac{I_c^+ - |I_c^-|}{I_c^+ + |I_c^-|}$ is given by $\eta \propto \varphi_0 \propto \mu_0 H_\parallel \cdot \sin(\theta_{MC})$. Here $\theta_{MC}$ is the relative angle between the polar axis (// $b$-axis) of WTe$_2$ and the applied $\mu_0 H_\parallel$[7,24–26]. The magneto-linearity ($\propto \mu_0 H_\parallel$) and the magneto-chirality [$\propto \sin(\theta_{MC})$] of $\eta$ are key measures of the intrinsic Josephson diode effect.

Figure 1c displays representative current-voltage ($I$–$V$) curves for a NbSe$_2$/$T_d$-WTe$_2$/NbSe$_2$ vdW JJ formed with a 23-nm-thick $T_d$-WTe$_2$ barrier. $I$–$V$ curves are shown when $\mu_0 H_\parallel = 0$ (black), +6 (orange) and −6 (green) mT are applied at a fixed $\theta_{MC} \approx 86.6°$ ($\theta = 45°$). These measurements are performed at 20 mK, far below the junction's $T_c \approx 5$ K. $\theta$ is the angle of $\mu_0 H_\parallel$ relative to an edge of the Si wafer on which the exfoliated layers were placed: the wafer had been cut into a rectangular shape to define a clear reference direction. Note that $\theta_{MC}$ in Fig. 1c is determined from comparison of the $\mu_0 H_\parallel$ angular dependence of $\eta$ of

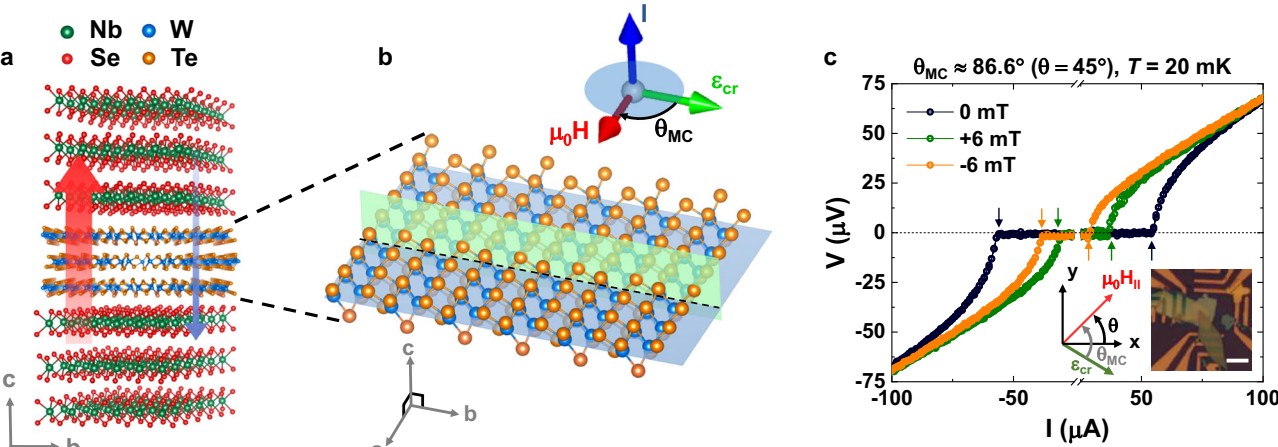

**Fig. 1 | Vertical rectified supercurrents in a van der Waals WTe$_2$ Josephson junction. a** Schematic of a NbSe$_2$/$T_d$-WTe$_2$/NbSe$_2$ van der Waals vertical Josephson junction (JJ). The combination of intrinsic inversion symmetry breaking within the $T_d$-WTe$_2$ barrier and time-reversal symmetry breaking by an applied magnetic field give rise to the (vertical) supercurrent non-reciprocity. **b** Schematic diagram of the crystal structure of WTe$_2$. Orange (blue) symbols represent the W (Te) atoms. $\theta_{MC}$ is

defined as the relative angle between the polar axis along $b$-axis, the mirror plane of $T_d$-WTe$_2$ (green plane), and the direction of the in-plane (IP) magnetic field $\mu_0 H_\parallel$ applied within the $a$-$b$ plane of $T_d$-WTe$_2$ (blue plane). **c** Current-voltage $I$–$V$ curves of a van der Waals JJ with a 23 nm thick $T_d$-WTe$_2$ barrier for $\mu_0 H_\parallel = 0$ (black), +6 (orange) and −6 (green) mT. Note that $\theta$ is the angle of $\mu_0 H_\parallel$ relative to an edge of the rectangular Si wafer on which the JJ was formed.

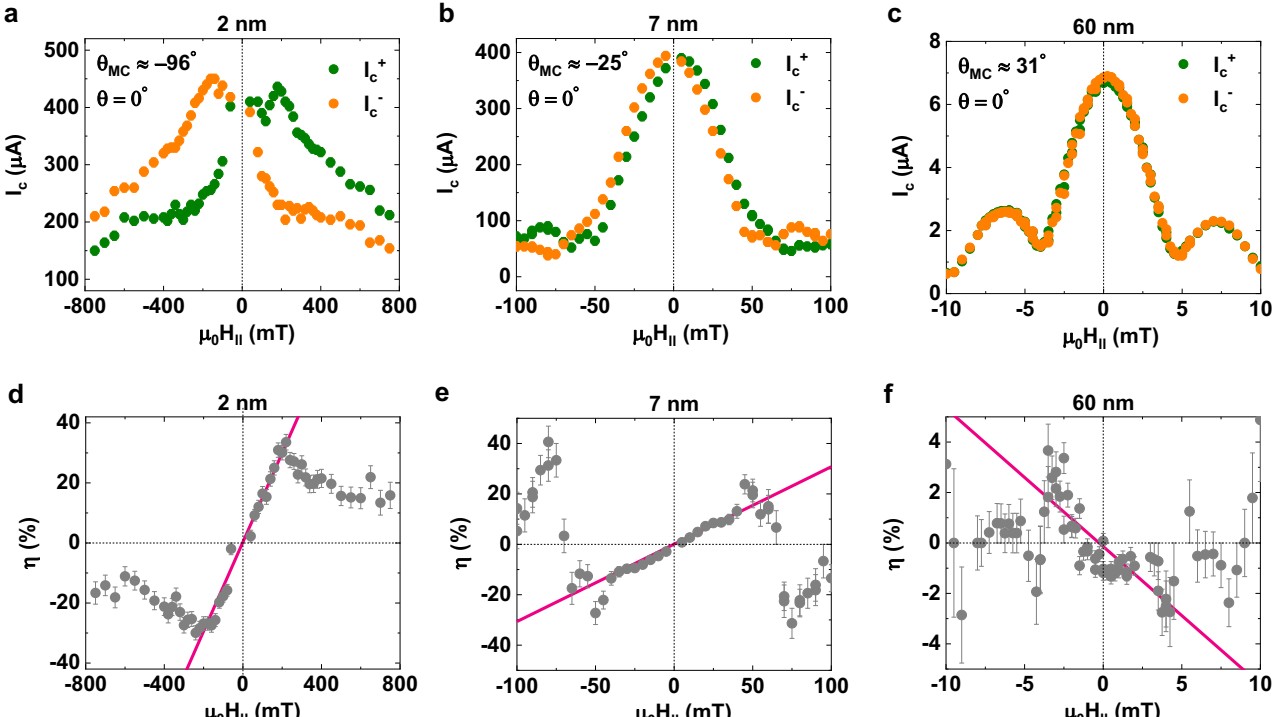

**Fig. 2 | Scaling of Josephson supercurrent non-reciprocity with magnetic field strength. a–c** Positive and negative Josephson critical current $I_c^+$ (green) and $|I_c^-|$ (orange) *versus* in-plane (IP) magnetic field $\mu_0 H_\parallel$ for **a** 2 nm, **b** 7 nm, and **c** 60 nm thick $T_d$-WTe$_2$ barriers in a NbSe$_2$/$T_d$-WTe$_2$/NbSe$_2$ Josephson junction. The measurement was conducted at $T = 200$ mK for the 2 nm and 7 nm thick junctions and $T = 20$ mK for the 60 nm thick junction, respectively. Note that the $I$–$V$ curves of

7 nm thick barrier JJs reveal no significant difference between 200 and 20 mK (Supplementary Information S7), as would be expected for the measurement temperature of $\leq 0.3\ T/T_c$ (Fig. 4b). **d–f** Josephson diode efficiency ($\eta = \frac{I_c^+ - |I_c^-|}{I_c^+ + |I_c^-|}$) as a function of $\mu_0 H_\parallel$ for **d** 2 nm, **e** 7 nm, and **f** 60 nm thick $T_d$-WTe$_2$ barriers. Note that the pink fits indicate a linear scaling behavior of the diode efficiency with $\mu_0 H_\parallel$ up to a critical field, above which the diode efficiency drops.

JJ with polarization-resolved Raman spectra (see Supplementary Information S3 for details). It is clear that a critical current asymmetry ($\triangle I_c = I_c^+ - |I_c^-| \neq 0$) is only developed when $\mu_0 H_\parallel$ is non-zero and that the polarity of $\triangle I_c$ is inverted when the direction of $\mu_0 H_\parallel$ is reversed. These features correspond to a Josephson diode effect. Note that since our $T_d$-WTe$_2$ barrier JJs show a weak hysteresis in the $I$–$V$ curves, indicating overdamped characteristics and the ignorable re-trapping, these can thus be the best example of experimental investigating the intrinsic Cooper-pair tunneling non-reciprocity by circumventing the unintentional contribution of quasiparticle tunneling[25] to the non-reciprocity.

The magneto-linearity of $\eta (\propto \mu_0 H_\parallel)$ is first checked by measuring how $I_c^+$ and $|I_c^-|$ vary with the strength of $\mu_0 H_\parallel$ for a fixed $\theta_{MC} \neq 0^\circ$ ($\theta = 0^\circ$). From a comparison of $|I_c(\mu_0 H_\parallel)|$ and $\eta(\mu_0 H_\parallel)$ for several values of the thickness of WTe$_2$ (2, 7, and 60 nm) (Fig. 2a–f), three notable features are revealed. First, for all the JJs, $\eta$ increases linearly with increasing $\mu_0 H_\parallel$ up to a certain critical field, above which $\eta$ starts to decay (Fig. 2d–f). Second, when this field coincides with the first-order minimum field of the Fraunhofer interference pattern, $\eta$ decays in a complex way with an accompanying sign change (Fig. 2b, c, e, and f). Third, the slope of the low-field $\eta(\mu_0 H_\parallel)$ data (Fig. 2d–f) depends critically on the orientation of the polar axis of the WTe$_2$ with respect to the applied $\mu_0 H_\parallel$. While the first and second features are qualitatively similar to previous studies on Al/2DEG/Al[7] and Nb/NiTe$_2$/Nb[13] lateral JJs, the third, signifying the magneto-chirality of $\eta [\propto \sin(\theta_{MC})]$, is a new finding from the NbSe$_2$/WTe$_2$/NbSe$_2$ vertical JJs. Note that no signature of fast oscillations, often attributed to topological edge states of WTe$_2$[27], are detected in measurements of the magnetic-field interference patterns for our vertical JJs (Fig. 2a–c). This is in agreement with theoretical considerations that topological edge states, that reside on the $a$-$b$ planes of WTe$_2$[28], may contribute to lateral transport[29,30] but not

to vertical transport. This indicates the crystallographic origin of the WTe$_2$ barrier for the magneto-chirality that we report here.

To confirm this distinctive magneto-chirality, the $\mu_0 H_\parallel$ angular dependence of $\eta$ is measured for each JJ. Note that this measurement is conducted in the low-field regime where the magneto-linearity of $\eta (\propto \mu_0 H_\parallel)$ holds (see Fig. 2d–f). As summarized in Fig. 3a–c, the measured $\eta(\theta)$ data are all well fitted by a sine function, suggesting a magneto-chiral origin of the Josephson diode effect in the vdW WTe$_2$ JJs. Furthermore, there exists a visible shift between the sine fits to the $\eta(\theta)$ data for different WTe$_2$ JJs. This indicates that the crystal structure of the WTe$_2$ barrier indeed governs the Josephson supercurrent non-reciprocity in our system. The magneto-chirality described above predicts $\eta$ to be maximized (minimized) when the applied $\mu_0 H_\parallel$ is aligned along the $a$-axis ($b$-axis). Note that in Fig. 3a–c, there exist non-zero constant offsets that are irrelevant to the intrinsic supercurrent rectification of magneto-chiral origin[12]. It is highly likely that such constant offsets arise from the asymmetric interface of the junctions as discussed previously[15]. Furthermore, a small out-of-plane component of the applied magnetic field due to a misalignment can, in principle, generate the field-angle-independent offset (Supplementary Information S9). To identify the $a$- and $b$-axis of each WTe$_2$ barrier in the fabricated junctions, angle-resolved polarized Raman spectroscopy was carried out: this technique can readily characterize the crystal orientation of low-dimensional materials[20,22]. Figure 3d–f, show the polarization angle dependent relative intensities of two distinct Raman peaks at ~160 cm$^{-1}$ and ~210 cm$^{-1}$. The extracted $a$- and $b$-axis of the WTe$_2$ from the Raman data (Fig. 3d–f, see Supplementary Information S3 for the detailed analysis) agree with those from the $\eta(\theta)$ data (Fig. 3a–c). These results strongly support an intrinsic-crystal-structure-reflected vertical Josephson diode effect. For completeness, we also fabricated and measured control vdW JJs, where the

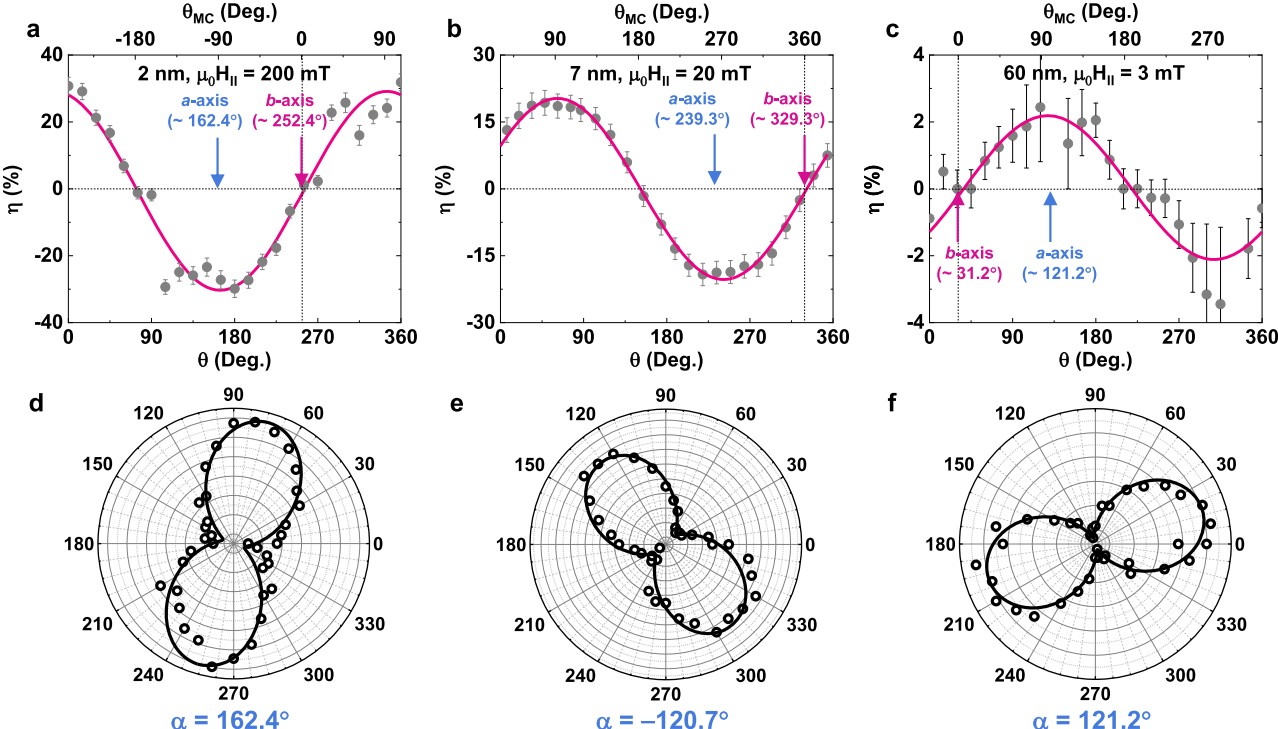

**Fig. 3 | Magneto-chiral angular dependence of the supercurrent non-reciprocity on the WTe$_2$ crystal structure. a–c** Field angle dependent Josephson diode efficiency ($\eta = \frac{I_c^+ - |I_c^-|}{I_c^+ + |I_c^-|}$) of NbSe$_2$/T$_d$-WTe$_2$/NbSe$_2$ junctions with **a** 2 nm, **b** 7 nm and **c** 60 nm thick $T_d$-WTe$_2$ barriers. Pink lines represent sine fits to the data. Top $x$-axis of the graphs show the $\theta_{MC}$ of each WTe$_2$ junctions. **d–f** Polarization angle dependent relative intensity of two distinct Raman peaks ( ~160 cm$^{-1}$ and ~210 cm$^{-1}$), from which the $a$- and $b$-axis of the WTe$_2$ can be determined. The $a$-axis and $b$-axis directions of the WTe$_2$ flake found in this way are indicated by arrows in (**a–c**).

non-centrosymmetric WTe$_2$ barrier is replaced with a MoTe$_2$ barrier that is locally centrosymmetric and may have a mixed $T_d$/1 $T'$ structure (see Supplementary Information S4)[31]. In addition, the effect of direct tunneling between the top and bottom superconducting NbSe$_2$ electrodes in JJs without any vdW-WTe$_2$ (Supplementary Information S6) were fabricated. None of the magneto-linearity and the magneto-chirality effects are found in these experiments (see Supplementary Information S4), thereby pinpointing the critical role of the Josephson barrier's crystal structure for the supercurrent non-reciprocity in our WTe$_2$ JJs. Based on these results, we can rule out other extrinsic possibilities for the supercurrent non-reciprocity in our WTe$_2$ JJs, e.g., pinning of Abrikosov vortices or a kinetic inductance contribution which have little to do with the crystal structure of the Josephson barrier (see Supplementary Information S8 for more discussions). In addition, one may expect the vdW gap formed at either side of the WTe$_2$ Josephson barrier to function as a tunnel barrier at the interface between NbSe$_2$ and WTe$_2$ flakes. As shown by the exponential decay of the characteristic voltage of WTe$_2$ JJs as a function of the thickness of the WTe$_2$ barrier (Supplementary Information S5), this is the case and so, our WTe$_2$ JJs need to be regarded as superconductor-insulator (vdW gap)-normal metal-insulator-superconductor (SINIS) junctions. This is in contrast with a previous study on lateral Nb/WTe$_2$/Nb JJs[32] that in the case of ballistic junctions with transparent/direct contacts, the barrier-thickness-dependent characteristic voltage is expected to reveal the 1/$L$ dependence in the long junction limit. Here $L$ is the lateral spacing of the neighboring Nb electrodes. Nevertheless, given the presence (absence) of crystal-structure-coupled magneto-chirality in WTe$_2$ (MoTe$_2$) JJs, it is unlikely that the crystal-structure-less vdW gaps play a significant role in producing such a characteristic magneto-chirality.

The proportionality of $\varphi_0$ to the barrier length is another characteristic feature of the intrinsic Josephson diode effect[24–26], so one can anticipate that the thicker the WTe$_2$ barrier, the greater the diode efficiency. To test this, we next investigate how the field-strength-

normalized diode efficiency $\eta^* = \frac{\eta}{\mu_0 H_\parallel}$, measured at $\theta_{MC} = 90°$, scales with the WTe$_2$ thickness (Fig. 4a). Note that since $\eta \propto \mu_0 H_\parallel$ in the low-field regime, $\eta^*$ ($= \frac{\eta}{\mu_0 H_\parallel}$) allows for a more quantitative comparison. As the WTe$_2$ thickness is increased from 2 to 23 nm, we find that $\eta^*$ increases linearly. This linear scaling behavior is, in fact, predicted theoretically for a ballistic JJ[24]. When the thickness of the WTe$_2$ is increased to 60 nm, which is larger than the coherence length (~30 nm, Supplementary Information S5) and the $c$-axis mean free path (~40 nm)[33], $\eta^*$ deviates from the linear thickness dependence. Given the distinctively different barrier-thickness dependence of $\varphi_0$ on whether the junction is in the ballistic or diffusive regime, such a deviation is likely related to a ballistic-to-diffusive (long-junction) transition[24–26]. As the $c$-axis mean free path is close to the $c$-axis coherence length of our WTe$_2$ JJs, one should also consider a short-to-long junction transition[32] *within* the ballistic limit (Supplementary Information S5). To clarify whether the non-monotonic thickness-dependent $\eta^*$ results from the ballistic-to-diffusive transition[24–26] or the short-to-long ballistic transition[32], further study is necessary in the future.

The temperature $T$ dependent evolution of $\eta^*$, gives additional information about the intrinsic Josephson diode properties. Figure 4b shows the $T/T_c$-dependent $\eta^*$ for several WTe$_2$ JJs. It is especially noteworthy that the $\eta^*(T/T_c)$ data can be well described by a $\sqrt{1-\frac{T}{T_c}}$ function for $\frac{T}{T_c} \gtrsim \frac{1}{3}$, as predicted in several earlier theoretical studies[24–26,34–36]. For $\frac{T}{T_c} < \frac{1}{3}$, $\eta^*$ tends to saturate, which is qualitatively similar to recent related experiments[7,11]. A very recent theory predicts a rather complicated $T$-dependence of the superconducting diode effect in diffusive Rashba-type SCs[35] as the low-$T$ limit diode efficiency is affected sensitively by structural disorder and impurity scattering. On the other hand, for Al/2DEG/Al ballistic JJs[7], the field-strength-dependent supercurrent non-reciprocity is explained by considering contributions from first and second order harmonics in the JJ CPR. If we apply this analysis

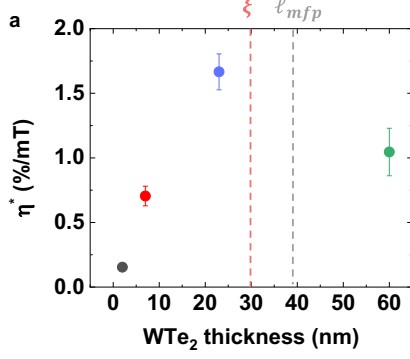

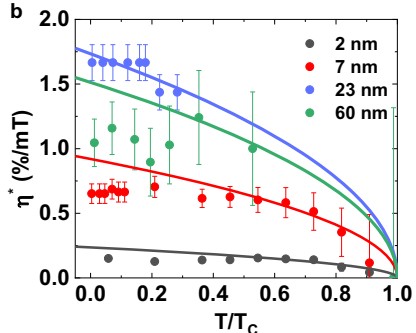

**Fig. 4 | Thickness and temperature dependent Josephson supercurrent non-reciprocity. a** Field-normalized Josephson diode efficiency $\eta^*$ ($= \frac{I_c^+ - |I_c^-|}{I_c^+ + |I_c^-|} / \mu_0 H_\parallel$), measured at $\theta_{MC} = 90°$ as a function of the $T_d$-WTe$_2$ barrier thickness. Vertical red and gray dashed lines correspond to the coherence length (Supplementary Information S4) and $c$-axis mean free path[22] of the $T_d$-WTe$_2$. **b** Normalized dependence of

$\eta^*$ as a function of normalized temperature $T/T_c$ of NbSe$_2$/$T_d$-WTe$_2$/NbSe$_2$ Josephson junctions with different $T_d$-WTe$_2$ barrier thicknesses (2, 7, 23 and 60 nm). The $T_c$ values of WTe$_2$ JJs are 5.5 K (2 nm and 7 nm), 5 K (23 nm) and 1.5 K (60 nm), respectively. Solid lines represent fits to the data of $\sqrt{1 - \frac{T}{T_c}}$.

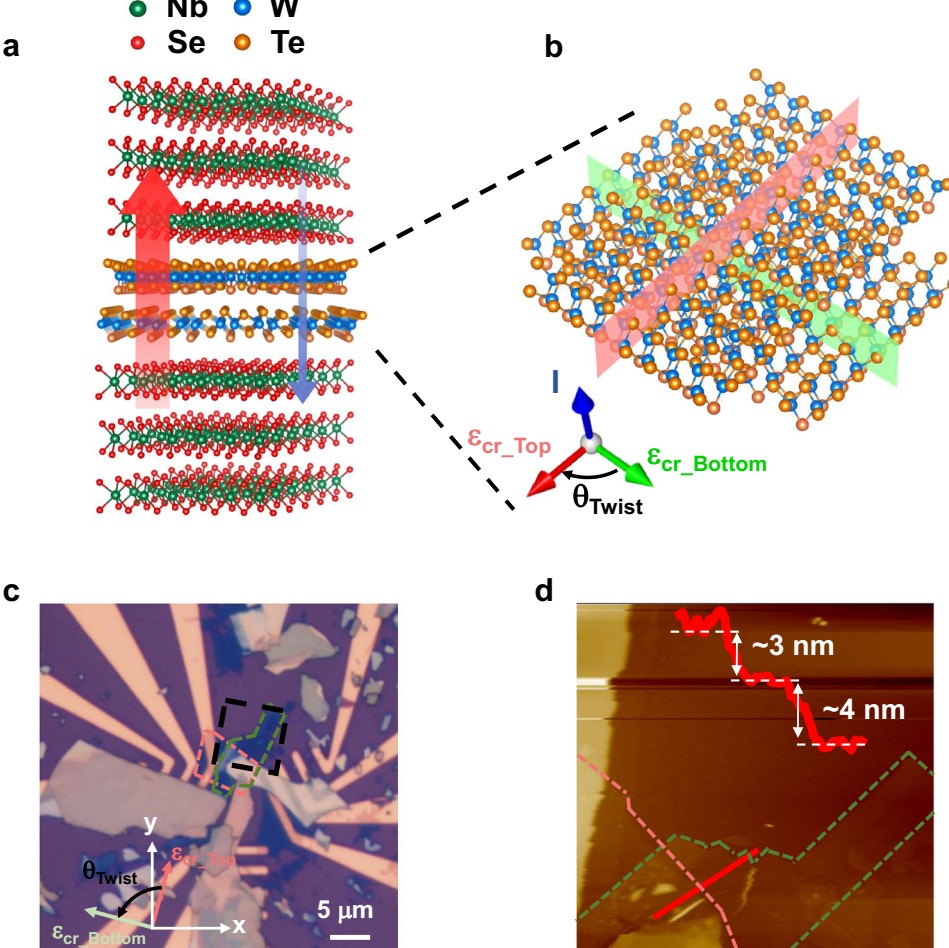

**Fig. 5 | Fabrication of twisted WTe$_2$ double-barrier JJ. a** Schematic diagram of a NbSe$_2$/twisted-WTe$_2$ double-barrier/NbSe$_2$ vdW vertical JJ. **b** magnified schematic of the crystal structure of the twisted-WTe$_2$ double-barrier. Notably, the vector addition of the internal crystal fields $\varepsilon_{cr\_Top}$ and $\varepsilon_{cr\_Bottom}$ of the top and bottom WTe$_2$ layers, respectively, with the twist angel $\theta_{twist}$ leads to a new artificial polar axis of the entire Josephson barrier. **c** Optical micrograph of the twisted-WTe$_2$

double-barrier JJ. The top (bottom) WTe$_2$ flake is marked by the red (green) dashed line. $\varepsilon_{cr\_Top}$ ($\varepsilon_{cr\_Bottom}$) is defined to be along the polar axis ($b$-axis) of the top (bottom) WTe$_2$. These axes are directly probed by angle-resolved polarized Raman spectroscopy, as depicted on the optical micrograph. The black dashed box represents the area scanned by AFM as shown in (**d**). **d** AFM image of respective WTe$_2$ barriers. Inset: depth profile along the red solid line.

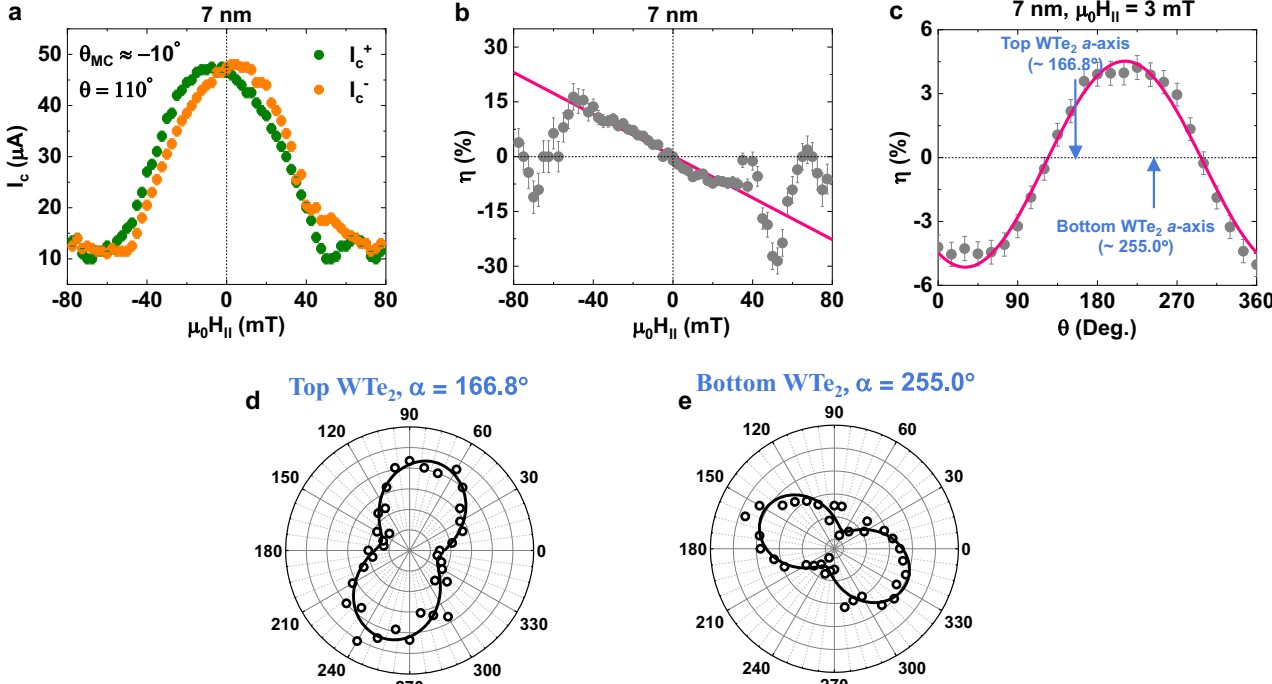

**Fig. 6 | Tuned magneto-chirality using the artificial polar axis of a twisted WTe$_2$ double-barrier JJ. a** Positive and negative Josephson critical current $I_c^+$ (green) and $|I_c^-|$ (orange) *versus* in-plane (IP) magnetic field $\mu_0 H_\parallel$ for twisted-WTe$_2$ double barrier JJ. Josephson diode efficiency $\eta (= \frac{I_c^+ - |I_c^-|}{I_c^+ + |I_c^-|})$ as a function of magnetic field strength (**b**) and angle (**c**) for the twisted WTe$_2$ double-barrier JJ. Polarization-angle-dependent relative intensity of two distinct *R*aman peaks (~160 cm$^{-1}$ and ~210 cm$^{-1}$),

from which one can determine the *a*- and *b*-axes of the top (**d**) and bottom (**e**) WTe$_2$ barriers. The measured *a*-axis directions of the top and bottom WTe$_2$ vdW layers are indicated by arrows in (**c**). As can be seen in (**c**) the magneto-chiral non-reciprocal supercurrents of the twisted-WTe$_2$ double barrier JJ are successfully controlled by twisting the top WTe$_2$ barrier relative to the bottom one.

to our $\eta^*(T/T_c)$ data, both harmonics seem to become constant at $\frac{T}{T_c} < \frac{1}{3}$, as is consistent with the calculation in Ref. 7

To further show the importance of our vertical JJ platform, that goes beyond recent studies[7–12], we have fabricated vertical JJs with a twisted WTe$_2$ double-barrier (Methods). In these twisted double-barrier junctions, *the vector addition of the internal crystal fields of the top and bottom WTe$_2$ layers*, respectively, with a twist angle between two layers leads to a new artificial polar axis of the entire Josephson barrier. Note that the twist angle $\theta_{twist}$ between two distinct WTe$_2$ layers uniquely forms an artificial polar axis of the entire Josephson barrier in a controllable manner (Fig. 5). In this sense, one can expect that the $\theta_{MC}$ is determined with respect to this new artificial polar axis. As shown in Fig. 6c, the diode efficiency of a twisted WTe$_2$ double-barrier JJ shows a visible sinusoidal behavior and the diode efficiency of the JJ goes to zero when the magnetic field is applied to the parallel direction of the vector addition of the internal crystal field of top and bottom WTe$_2$ layers. This indicates that the magneto-chirality of twisted-double barrier JJ can be determined by the artificial polar axis of the barrier, demonstrating a tunable magneto-chirality via twist-angle engineering and opens an avenue for the development of twistable[36] active Josephson diodes.

We have carefully fabricated vdW vertical JJs with an inherently inversion symmetry breaking $T_d$-WTe$_2$ barrier. This has allowed us to unambiguously demonstrate the intrinsic origin of the Josephson non-reciprocity that we observe in these vertical JJs. We clearly demonstrate that the crystal structure of the $T_d$-WTe$_2$ Josephson barrier governs the overall properties of the Josephson diode effect, which has not been previously been shown. We have also shown that whether or not the magneto-linearity, the magneto-chirality and the thickness and temperature-scaleable diode efficiency coexist is a key criterion in distinguishing intrinsic and extrinsic mechanisms. We believe that our results have definitely shown the critical role of the barrier in

the realization of rectified Josephson supercurrents in vdW heterostructures[15] and provides the first demonstration of twist engineering of the magneto-chirality. Our approach can be extended to other low-dimensional, low-symmetric and twisted vdW systems[37–39] for accelerating the development of two-dimensional superconducting devices and circuits.

## Methods
### Device fabrication
The NbSe$_2$/WTe$_2$/NbSe$_2$ van der Waals (vdW) heterostructures (Fig. S1a–d) were formed using standard dry transfer techniques. All the needed processes, including mechanical exfoliation, pick-up and dry transfer of the vdW flakes, were performed inside a nitrogen gas filled glovebox to prevent the flakes from oxidizing in ambient air. First, NbSe$_2$ ($\geq$ 10 nm thick) and WTe$_2$ (2–60 nm thick) flakes were exfoliated on to a 300-nm-thick SiO$_2$/p + + Si substrate. Each of the flakes needed to form the heterostructure was chosen by optical microscopy examination. The chosen flakes were picked up sequentially using a polycarbonate (PC) film coated dome-shaped polydimethylsiloxane (PDMS) stamp. The flakes were aligned one on top of each other and them the entire stack was released and then placed on top of a set of pre-patterned gold electrodes on a second 300-nm-thick SiO$_2$/p + + Si substrate. The release process was performed at 200 C°, after which the entire stack in the Si substrate was immersed in a chloroform solution to remove any PC residue. Using this fabrication process flow, we also prepared three distinct types of JJs: (1) a control vdW JJ, in which the non-centrosymmetric WTe$_2$ barrier[20,22,23] is replaced by a MoTe$_2$ barrier[31] that may have a *mixed* $T_d/1\,T'$ structure and is to a certain extent locally centrosymmetric at low temperatures (Supplementary Information S4), (2) a reference vdW JJ with *no* WTe$_2$ barrier (Supplementary Information S6), and (3) a twisted WTe$_2$ double-barrier JJ (Figs. 5–6), where one WTe$_2$ is

twisted in-plane relative to the other $WTe_2$. For the formation of the twisted $WTe_2$ double-barriers, we first identified the crystal orientation of two chosen $WTe_2$ flakes through their elongated shapes[20] (which tends to be along the $a$-axis), and then carefully twisted and stacked one on top of the other to realize $\theta_{twist} \approx 90°$. After completing the transport measurements, we finally confirmed the respective crystal orientations of the top and bottom $WTe_2$ flakes by a means of angle-resolved polarized Raman spectroscopy.

## Electrical measurements

The current-voltage ($I$–$V$) curves of the fabricated $NbSe_2/WTe_2/NbSe_2$ vdW vertical Josephson junctions (JJs) were measured in a BlueFors dilution refrigerator or a Quantum Design Physical Property Measurement System with base temperatures of ~20 mK and ~1.8 K, respectively. The four-point $I$–$V$ measurements were carried out using a Keithley 6221 current source and a Keithley 2182 A nanovoltmeter. We determined the critical current $I_c$ values at the point where $V(I) \approx$ 1 μV. An external in-plane magnetic field ($\mu_0 H_\parallel$), that was needed to break the time-reversal symmetry, was applied within the $a$-$b$ plane of the $WTe_2$ single crystalline barrier. Note that $\theta$ is the angle of $\mu_0 H_\parallel$ relative to one edge of the Si wafer on which the exfoliated layer stack was placed: the wafer had been purposely cut into a rectangular shape to define a clear reference direction. The typical Josephson penetration depth[40] of our junctions (of at least a few μm) is much larger than the $WTe_2$ thickness, so we can exclude orbital magnetic field effects (or Meissner demagnetizing supercurrents) as a possible extrinsic source for of the Josephson diode effect in our JJs. We measured the field strength dependent diode signals (Fig. 2a–c) by applying $\mu_0 H_\parallel$ along a given $\theta$ direction. The angle dependent diode signals (Fig. 3a–c) were measured at a constant $\mu_0 H_\parallel$ using a vector field magnet.

## Atomic force microscopy

To characterize the thickness of each $WTe_2$ barrier, we conducted atomic force microscopy (AFM) measurements on the fabricated vdW vertical JJs. The measured AFM image and height of each $WTe_2$ barrier layer are shown together with the corresponding optical micrograph of the JJs in Fig. S1.

## Data availability

The data that support the findings of this study are provided in main text and the Supplementary Information. The original data available from the corresponding authors upon request.

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

## Acknowledgements
SSPP acknowledges funding from the Deutsche Forschungsgemeinschaft (DFG, German Research Foundation)—project no. 443406107, Priority Programme (SPP) 2244. K.-R.J. acknowledges that this work was supported by Chung-Ang University Research Grant in 2022.

## Author contributions
J.-K.K. and S.S.P.P. conceived of and designed the experiments with the help of K.-R.J. J.-K.K. fabricated the van der Waals Josephson junctions and carried out the transport measurements with the help of K.-R.J., P.K.S. and J.J. J.-K.K. performed the Raman measurement with the help of C.K. and G.W. J.-K.K analyzed the data with the guidance of K.-R.J. J.-K.K., K.-R.J and S.S.P.P. wrote the manuscript with input from all the other authors.

## Funding

## Competing interests
The authors declare no competing interests.
