## [Peer Review File · Nature Communications]

Intrinsic supercurrent non-reciprocity coupled to the crystal structure of a van der Waals Josephson barrierReviewers' Comments:

Reviewer #1:

Remarks to the Author:

This paper presents interesting data on the field dependence of the critical current in vertical SNS junctions where the normal part is a WTe₂ multi-layered crystal whose crystalline structure is not centro-symmetric is imbedded between NbSe₂ superconducting electrodes. The authors observe a convincing non-reciprocity in the critical current with respect to the sign of the current and in-plane applied magnetic field. This asymmetry depends on the orientation of the magnetic field with respect to the crystal orientation as well as the thickness of the junction.

These results point towards the intrinsic origin of this effect of non reciprocity and deserve publication .

However the presentation of the data suffers from some inconsistencies as well as missing important information on the samples investigated as detailed below:

1-Normal state transport in the samples and nature of the Josephson junctions investigated.

An important parameter for the understanding of SNS Josephson junction is their normal state resistance. This simple parameter is impossible to deduce from the sparse and confusing information given in the paper and supplementary.

From Fig.1 one would get for the 23nm thick sample R_n of the order of 1K Ohm yielding a $R_n I_c$ product of 50mV which is completely unphysical for the NbSe₂ superconducting gap. On the other hand the values of the $R_n I_c$ product given in Fig S5 is 0.5 mV which is more reasonable. This would yield instead $R_n=10$ Ohm! The nature of the transport is also not clear.

Authors use the term barrier which is adequate for insulators. However this would imply an exponential increase of the normal states resistance with respect to the thickness of the Td-WTe₂ flake which does not agree with the semi metallic dependence observed in ref.29 as well as the metallic character observed in 20nm or more thick Td-WTe₂ samples by Zhang et al. PRB 104, 165126 (2021). Moreover authors claim that transport is ballistic through the junction with a mean free path of the order of 40nm which is indeed observed in other work. For a metallic sample this would yield R_n and I_c both nearly length independent below 40nm which is not consistent with the data obtained. One cannot exclude the existence of barriers at the NbSe₂ WTe₂ interface (see so called SINIS junctions discussed extensively in the literature by Golubov, Kupryanov etc..) but this is not at all mentioned by the authors. Note that such a double barrier model could also explain the exponential decay V_c as a function of the WTe₂ sample shown in Fig.S5. This is maybe what authors have in mind but this is not explained. Note also that a $1/L$ behaviour would fit the data in Fig.S5 as well.

2-Measurement of the critical current . Authors do not explain the way they measure the critical current of their Junctions . The data obtained is particularly noisy in spite of the rather large values of the critical current.

Can the author explain the poor quality of their data? In particular the data on MoTe₂ (Inversion-symmetric sample) is not really convincing due to the large amount of noise .

3-Discussion of the possible kinetic inductance effects of the electrodes.

The superconducting electrodes are made from thin flakes (10nm) of NbSe₂. It would be important to estimate the kinetic inductance of these electrodes. It is indeed well known that the presence of a large inductance in series with a Josephson Junction can also generate magneto-chiral asymmetries. See for example Schonle et al. Scientific Reports , 9:1987 (2019).

All these points need to be discussed and clarified before publication.

Reviewer #2:

Remarks to the Author:

The authors present a systematic study exploring the superconducting diode effect (SDE) in vertically stacked NbSe₂/WTe₂ heterostructures (as well as control samples using MoTe₂, twisted WTe₂ double

barrier and vertically stacked NbSe₂). They find that non-reciprocity in the devices emerges under the application of magnetic field. From here, the authors demonstrate the magneto-linearity and magneto-chiral aspects of the non-reciprocity, which they argue is born from the crystal asymmetry of the device.

I find the study very thorough in discussing the layer dependence of the NbSe₂/WTe₂ structures as well as providing several controls. I find their argument convincing and suitable for publication in Nature Communications after the following minor points are addressed:

1) The authors do not make any mention of the possibility of vortices causing a SDE in their devices (see, for instance, Sundaresh et. al Nat. Comm. 2023). This interpretation would be consistent with the thickness dependence observed in their devices as the increasing critical currents at lower thicknesses would drive higher non-reciprocities due to pinned vortices. However, this effect can be disregarded by noting the absence of a SDE in the control device utilizing MoTe₂. Here, the MoTe₂ device has comparable critical currents to the 23 nm device, which does exhibit non-reciprocity. This contrast shows that trapped vortices are not a likely cause of the non-reciprocity and gives credence to the author's explanation. (To some degree this is also verified by the NbSe₂/NbSe₂ device with no intervening tunnel barrier, however this situation is less analogous to the WTe₂ structures due to the absence of a tunnel barrier). I suggest the authors include this discussion to make this rationale more accessible to the reader.

2) The authors do not mention the operating temperature for Figure 1 other than to indicate it is below the critical temperature. Mentioning the temperature in Figure 1 is important to best assess the junction damping as it roughly informs the reader of the dominant activation regime they will be in (macroscopic quantum tunneling, thermal activation, or phase diffusion). This allows one to properly evaluate if the lack of hysteresis in the junction is due to junction damping or a symmetry in the IV as the device enters the phase diffusion regime.

3) While the authors discuss developments in studying the SDE from the materials level in great detail, they overlook a substantial portion of the efforts examining supercurrent non-reciprocity at the device level. To further encapsulate the broader history of the field, it may be prudent to include a brief discussion of these efforts. See for instance: Goldman et al, PRL 1967; Fulton et. al. Phys Rev B 1972; Peterson et. al. J. Appl. Phys 1979; Frattini et. al. Appl. Phys. Lett 2017; Lyu et al. Nat. Comm 2021; Davydova et. al. Sci. Adv. 2022; Chiles et. al. Nano Lett 2023; Gupta et. al. Nat. Comm 2023.

Reviewer #3:

Remarks to the Author:

The paper by Kim et al. describes the observation of the superconducting diode effect (SDE) in a vertical Josephson junction (JJ) made by NbSe₂ electrodes and a WTe₂ barrier. The field of SDE has been very popular in the last few years, with many papers published every month, and there have already been papers on SDE in vertical JJs. This poses the question of novelty, and unfortunately the paper is not very clear about the novelty of these data. Thus, either the authors succeed in improving the description of the novelty of this paper in a revised version, or the manuscript might be better suited for a journal like Communications Physics.

Besides this, there is a series of comments that the authors might want to consider before resubmission of the revised version of the manuscript:

1) The introduction is in general too brief and does not provide an overview of the relevant literature, in particular on SDE in Josephson junctions.

2) The paper at least twice states that $\eta \propto \sin(\theta_{MC})$, but there is no plot to show this. Besides, on page 4 it is claimed that $\eta \propto \sin(\theta_{MC})$ is a new finding. However, a similar sine dependence of the magneto-chiral anisotropy has been reported before by Baumgartner et al., J. Phys.: Condens. Matter 34 (2022) 154005.

3) What is the influence of a possible small vertical component of the applied in-plane magnetic field due to a misalignment? Could this explain some of the findings?

4) What is the temperature for the data shown in Fig. 1c? The transition from the superconductive to the normal state is rather rounded, which might indicate that these curves were not taken at base temperature. Maybe the junction is overdamped because of the high temperature? In brief, the authors should show IV-curves at base temperature.

5) The order of the presentation in the paper could be improved. Fig. 2 gives and discussed values for θ_{MC} , but only in Fig. 3 it is explained how these are obtained.

6) Fig. 2 compares junctions of different barrier thickness. However, besides barrier thickness, also temperature is changed. It would be better to show a consistent dataset in which only one parameter at a time is changed.

7) Fig. 4, give the values of T_c .

8) It is surprising that the possibly most interesting dataset (on the twisted double-barrier) is hidden in the Extended Data. Why? In any case, the discussion of this sample should be extended, to provide sufficient information. Referring to point 2, what is the angular dependence of the SDE efficiency expected for the twisted bilayer case? Which function was used to fit the data in Extended Fig. 2c, and what is the physical model behind the fit function? It seems to be not a single sine function, and appears to be not even periodic with 360deg? A more detailed discussion is required.

9) Besides, when was the α of the two barrier layers measured? Before or after the assembly of the single layers into the stack?

10) SI chapter S6: if there is no barrier, why does the device behave as if there was a barrier? What is acting as a barrier here? Is the NbSe₂ oxidized?

Detailed response to referee's comments.

In the following, we provide detailed point-by-point responses to the referees' comments (referees' comments in blue; our responses in black).

A. Response to the First Referee:

This paper presents interesting data on the field dependence of the critical current in vertical SNS junctions where the normal part is a WTe₂ multi-layered crystal whose crystalline structure is not centro-symmetric is imbedded between NbSe₂ superconducting electrodes. The authors observe a convincing non-reciprocity in the critical current with respect to the sign of the current and in-plane applied magnetic field. This asymmetry depends on the orientation on the magnetic field with respect to the crystal orientation as well as the thickness of the junction. These results point towards the intrinsic origin of this effect of non reciprocity and deserve publication. However the presentation of the data suffers from some inconsistencies as well as missing important information on the samples investigated as detailed below:

We appreciate the referee for his/her positive evaluation of our paper for publication in Nature Communications.

Specific comments:

1. Normal state transport in the samples and nature of the Josephson junctions investigated. An important parameter for the understanding of SNS Josephson junction is their normal state resistance. This simple parameter is impossible to deduce from the sparse and confusing information given in the paper and supplementary. From Fig.1 one would get for the 23nm thick sample R_n of the order of 1K Ohm yielding a $R_n I_c$ product of 50mV which is completely unphysical for the NbSe₂ superconducting gap. On the other hand the values of the $R_n I_c$ product given in Fig S5 is 0.5 mV which is more reasonable. This would yield instead $R_n=10$ Ohm! The nature of the transport is also not clear. Authors use the term barrier which is adequate for insulators. However this would imply an exponential increase of the normal states resistance with respect to the thickness of the Td-WTe₂ flake which does not agree with the semi metallic dependence observed in ref.29 as well as the metallic character observed in 20nm or more thick Td-WTe₂ samples by Zhang et al. PRB 104, 165126 (2021). Moreover authors claim that transport is ballistic through the junction with a mean free path of the order of 40nm which is indeed observed in other work. For a metallic sample this would yield R_n and I_c both nearly length independent below 40nm which is not consistent with the data obtained. One cannot exclude the existence of barriers at the NbSe₂ WTe₂ interface (see so called SINIS junctions discussed extensively in the literature by Golubov, Kupryanov etc..) but this is not at all mentioned by the authors. Note that such a double barrier model could also explain the exponential decay V_c as a function of the WTe₂ sample shown in Fig. S5. This is maybe what authors have in mind but this is not explained. Note also that a $1/L$ behaviour would fit the data in Fig.S5 as well. With respect to the 2 nm thick sample: I am a bit surprised that this device follows the same trend as the others, since at 2 nm one may expect direct tunneling of Cooper pairs to take place between the superconducting leads. The authors may want to comment on this.

We thank the reviewer’s careful reading of our paper. The y-axis unit of Fig. 1c was wrongly typed and should be μV (not mV). We corrected this typo in our revised manuscript.

Regarding the discussion on SINIS-like junctions, we agree with the reviewer’s comment. Van der Waals (vdW) gaps do exist at the interface between the WTe₂ barrier and the NbSe₂ electrodes and thus they can act as an insulating barrier. We have added a brief discussion of how the vdW gaps influence the supercurrent properties of our metallic JJs in the revised manuscript (page 5).

As the reviewer points out, direct tunneling between two adjacent superconducting leads may play a role in the transport properties of JJs, in particular, for those with atomically thin barriers. In the case of the 2-nm-thick WTe₂ barrier, we expect two different charge-transport mechanisms, 1) direct tunneling (of charge) from one to the other superconducting lead and 2) ballistic transport [Nat. Nanotechnol. **17**, 39 (2022)] through Andreev bound states (ABS) formed in the WTe₂ barrier. Given that the 2-nm-thick WTe₂ barrier junction clearly reveals magneto-chiral characteristics *with reference to its polar axis*, one can conclude the latter ballistic transport dominates over the former direct tunneling, which is expected to be independent of the crystal structure of WTe₂.

Furthermore, none of the magneto-linearity and the magneto-chirality are detected in these WTe₂-absent JJs (Fig. R1 (same as Fig. S6 in the original Supplementary Information), as would be expected for the vdW gap acting as a *crystal-structure-less* Josephson barrier. This result reinforces our discussion about the 2-nm-thick WTe barrier JJ in the paragraph.

Figure R1. **a**, Positive and negative Josephson critical current I_c^+ (green) and $|I_c^-|$ (orange) *versus* in-plane magnetic field $\mu_0 H_{||}$ for a WTe₂-absent JJ. All the measurements were conducted at $T = 2$ K. Josephson diode efficiency $\eta (= \frac{I_c^+ - |I_c^-|}{I_c^+ + |I_c^-|})$ as a function of magnetic field strength (**b**) and angle (**c**) for the WTe₂-absent JJ. We note that the diode efficiency (η) $<$ few % for the WTe₂-absent JJ is remarkably small as compared with the WTe₂-present JJs and none of the magneto-linearity and the magneto-chirality are evident in the WTe₂-absent JJ.

2. Measurement of the critical current. Authors do not explain the way they measure the critical current of their Junctions. The data obtained is particularly noisy in spite of the rather large values of the critical current. Can the author explain the poor quality of their data? In particular the data on MoTe₂ (Inversion- symmetric sample) is not really convincing due to the large amount of noise. The inset to Figure 1c is supposed to show the device, but it seems to show the electrodes instead.

To reflect this comment, we have specified how we determined the critical current of our junctions in the revised manuscript (Methods).

To improve the quality of our data, we have re-measured I - V Josephson curves, in particular, on the MoTe₂ control device more than 3 times and averaged the measured multiple curves to define the I_c values more accurately. Note that for fair comparison with other data, we focus our analysis on a low magnetic field region, in which the magneto-linearity of η holds (Fig. R2). From the refined data, one can be assured that no magneto-linearity and magneto-chirality are present for the MoTe₂ control device.

Fig. R2. **a**, Optical micrograph of 1T'-MoTe₂ Josephson junctions. Scale bar is 5 μm . Note that unlike the orthorhombic phase T_d -WTe₂, the monoclinic phase 1T'-MoTe₂ is centrosymmetric. **b**, Positive and negative Josephson critical current I_c^+ (green) and $|I_c^-|$ (orange) versus in-plane magnetic field $\mu_0 H_{\parallel}$ for a 12 nm thick 1T'-MoTe₂ barrier in a NbSe₂/1T'-MoTe₂/NbSe₂ Josephson junction. The measurement was conducted at $T = 20$ mK. Josephson diode efficiency ($\eta = \frac{I_c^+ - |I_c^-|}{I_c^+ + |I_c^-|}$) as a function of magnetic field strength (**c**) and angle (**d**) for the NbSe₂/1T'-MoTe₂/NbSe₂ Josephson junction.

3. Discussion of the possible kinetic inductance effects of the electrodes. The superconducting electrodes are made from thin flakes (10nm) of NbSe₂. It would be important to estimate the kinetic inductance of these electrodes. It is indeed well known that the presence of a large

inductance in series with a Josephson Junction can also generate magneto-chiral asymmetries. See for example Schonle et al. *Scientific Reports*, 9:1987 (2019). Figure 1e (supplementary), the dashed rectangle does not appear to overlap the AFM image.

We would first like to emphasize that the $\mu_0 H_{\parallel}$ angular dependence of η of WTe₂ Josephson junctions is intimately coupled to the crystal orientation of each WTe₂ barrier. So, we do not believe that the kinetic inductance (of extrinsic effects) is responsible for the magneto-chirality of WTe₂ JJs. The absence magneto-chirality in control devices (WTe₂-absent and *IT'*-MoTe₂ barrier JJs) decisively supports our intrinsic picture.

The kinetic inductance $L_K \approx \mu_0 \frac{\lambda^2 l}{wt}$ of thin NbSe₂ flakes in WTe₂ JJs is estimated to be ~33 pH for zero current and at zero temperature (where L_K is the smallest). Here, μ_0 is vacuum permeability and λ is the London penetration depth of bulk NbSe₂ (230 nm) [*J. Low Temp. Phys.* **11**, 421 (1973)]. l , w and t are the length, width and thickness of NbSe₂ electrodes, respectively. The estimated L_K value of our junctions is at least one order of magnitude smaller than that in a previous study [*Scientific Reports*, 9:1987 (2019)]. Furthermore, our measurement setup under application of in-plane magnetic fields differs from their one where supercurrent non-reciprocity appears even *in the absence of magnetic fields (i.e. no need for breaking the time-reversal symmetry)*. As a result, we can conclude that the kinetic inductance contribution to the supercurrent non-reciprocity of our WTe₂ JJs is insignificant.

Nevertheless, to reflect reviewer's comment, we have added a concise discussion on the kinetic inductance contribution to our data in the revised manuscript (Supplementary Section S8).

We have provided the correct micrograph (Fig. 1c) and AFM image (Fig. 1e) in the revised manuscript.

B. Response to the Second Referee:

The authors present a systematic study exploring the superconducting diode effect (SDE) in vertically stacked NbSe₂/WTe₂ heterostructures (as well as control samples using MoTe₂, twisted WTe₂ double barrier and vertically stacked NbSe₂). They find that non-reciprocity in the devices emerges under the application of magnetic field. From here, the authors demonstrate the magneto-linearity and magneto-chiral aspects of the non-reciprocity, which they argue is born from the crystal asymmetry of the device.

I find the study very thorough in discussing the layer dependence of the NbSe₂/WTe₂ structures as well as providing several controls. I find their argument convincing and suitable for publication in *Nature Communications* after the following minor points are addressed:

We appreciate the referee for his/her recommendation of our manuscript for publication in *Nature Communications*.

1. The authors do not make any mention of the possibility of vortices causing a SDE in their devices (see, for instance, Sundaresh et. al *Nat. Comm.* 2023). This interpretation would be consistent

with the thickness dependence observed in their devices as the increasing critical currents at lower thicknesses would drive higher non-reciprocities due to pinned vortices. However, this effect can be disregarded by noting the absence of a SDE in the control device utilizing MoTe₂. Here, the MoTe₂ device has comparable critical currents to the 23 nm device, which does exhibit non-reciprocity. This contrast shows that trapped vortices are not a likely cause of the non-reciprocity and gives credence to the author's explanation (To some degree this is also verified by the NbSe₂/NbSe₂ device with no intervening tunnel barrier, however this situation is less analogous to the WTe₂ structures due to the absence of a tunnel barrier). I suggest the authors include this discussion to make this rationale more accessible to the reader.

We appreciate the reviewer's valuable comment. We have added the suggested discussion that pinning of Abrikosov vortices is an unlikely origin of the supercurrent non-reciprocity for our WTe₂ Josephson junctions in the revised manuscript (page 5).

2. The authors do not mention the operating temperature for Figure 1 other than to indicate it is below the critical temperature. Mentioning the temperature in Figure 1 is important to best assess the junction damping as it roughly informs the reader of the dominant activation regime they will be in (macroscopic quantum tunneling, thermal activation, or phase diffusion). This allows one to properly evaluate if the lack of hysteresis in the junction is due to junction damping or a symmetry in the IV as the device enters the phase diffusion regime.

As the reviewer suggested, we have added the operating temperature in Fig. 1 of our revised manuscript.

3. While the authors discuss developments in studying the SDE from the materials level in great detail, they overlook a substantial portion of the efforts examining supercurrent non-reciprocity at the device level. To further encapsulate the broader history of the field, it may be prudent to include a brief discussion of these efforts. See for instance: Goldman et al, PRL 1967; Fulton et. al. Phys Rev B 1972; Peterson et. al. J. Appl. Phys 1979; Frattini et. al. Appl. Phys. Lett 2017; Lyu A. Nat. Comm 2021; Davydova et. al. Sci. Adv. 2022; Chiles et. al. Nano Lett 2023; Gupta et. al. Nat. Comm 2023.

We thank the referee for this comment. We have added a brief discussion on how supercurrent non-reciprocity at the device level has progressed in our revised manuscript.

C. Response to the third Referee:

The paper by Kim et al. describes the observation of the superconducting diode effect (SDE) in a vertical Josephson junction (JJ) made by NbSe₂ electrodes and a WTe₂ barrier. The field of SDE has been very popular in the last few years, with many papers published every month, and there have already been papers on SDE in vertical JJs. This poses the question of novelty, and unfortunately the paper is not very clear about the novelty of these data. Thus, either the authors succeed in improving the description of the novelty of this paper in a revised version, or the manuscript might be better suited for a journal like Communications Physics.

We thank the reviewer for his/her rigorous evaluation of our manuscript. We strongly are of the opinion that our results are novel and address the very important question as to whether or not there can be an intrinsic JDE. There clearly are cases where a JDE or SDE has an extrinsic origin e.g. from geometrical asymmetries in the junction. However, our study clearly shows that there are cases, as here, where the origin of the JDE effect is intrinsic. In particular, our study is the first to demonstrate a clear connection between the JDE effect and the crystal symmetry of the JJ barrier. There have been few studies on vertical JJs. We show by using a vertical JJ the field that we apply to break TRS can be aligned along different directions with respect to the polar axis in our non-centrosymmetric WTe₂ barrier. This experiment has not been carried out before. Thus, we believe our studies are an important and novel contribution to the field of intrinsic JDE and SDE effects. We have added a more discussion about the novelty of our studies to the intro/conclusions of our revised manuscript. We also would like to point out that we show clearly how the magnitude of the JDE effect (i.e. the magnitude of the non-reciprocity per unit field for small fields) increases with the thickness of the WTe₂ barrier to a maximum thickness of approximately ~23 nm which is another novel finding.

Besides this, there is a series of comments that the authors might want to consider before resubmission of the revised version of the manuscript:

1. The introduction is in general too brief and does not provide an overview of the relevant literature, in particular on SDE in Josephson junctions.

To reflect this comment, we have expanded the introduction section of our manuscript by overviewing the relevant recent studies.

2. The paper at least twice states that $\eta \propto \sin(\theta_{MC})$, but there is no plot to show this. Besides, on page 4 it is claimed that $\eta \propto \sin(\theta_{MC})$ is a new finding. However, a similar sine dependence of the magneto-chiral anisotropy has been reported before by Baumgartner et al., J. Phys.: Condens. Matter 34 (2022) 154005.

We appreciate the referee's comments but the plot of $\eta \propto \sin(\theta_{MC})$ is already clearly shown in Fig. 3a-c (please see pink curves that show the sinusoidal dependence of η on θ). However, in response to the reviewer's comment, we have added a new x -axis for θ_{MC} in top of Fig. 3a-c to unambiguously show $\eta \propto \sin(\theta_{MC})$ in the graph, as shown in Fig. R3.

In the previous study by Baumgartner et al. referenced by the referee, the magneto-chirality relies on the angle between the direction of an *interface* inversion symmetry breaking and the applied magnetic field. However, our study demonstrates a new magneto-chirality effect that depends on the angle between the direction of *bulk* inversion symmetry breaking and the applied magnetic field. This is our new finding and is a distinctively different result when compared with the previous study where the *interface* inversion symmetry breaking provide a limited degree of magneto-chirality engineering.

Fig. 3. Magneto-chiral angular dependence of the supercurrent non-reciprocity on the WTe₂ crystal structure. **a, b, c,** Field angle dependent Josephson diode efficiency ($\eta = \frac{I_C^+ - |I_C^-|}{I_C^+ + |I_C^-|}$) of NbSe₂/T_d-WTe₂/NbSe₂ junctions with **(a)** 2 nm, **(b)** 7 nm and **(c)** 60 nm thick T_d-WTe₂ barriers. Pink lines represent sine fits to the data. Top x-axis of graphs show the θ_{MC} of each WTe₂ junctions. **d, e, f,** Polarization angle dependent relative intensity of two distinct Raman peaks (~ 160 cm⁻¹ and ~ 210 cm⁻¹), from which the *a*- and *b*-axis of the WTe₂ can be determined. The *a*- and *b*-axis directions of the WTe₂ flake found in this way are indicated by arrows in **a, b, c.**

3. What is the influence of a possible small vertical component of the applied in-plane magnetic field due to a misalignment? Could this explain some of the findings?

Although there exists a possibility for the existence of a non-zero vertical component of applied in-plane magnetic fields due to few-degrees misalignment, it cannot explain the crystal-orientation-coupled supercurrent non-reciprocity of our WTe₂ JJs. This is because the non-zero vertical (misalignment) field is expected to *be independent of* rotating the in-plane magnetic field with reference to the polar axis of WTe₂. Thus, any small vertical misalignment cannot explain our data. We would like to re-emphasize that in 4 distinct junctions where we carried the WTe₂ thickness we demonstrated clearly from polarized Raman studies after we had carried out our studies of the dependence of the η on the in-plane field angle that η was clearly associated with the polar direction within the WTe₂ plane. In these 4 cases the orientation of this polar direction was different with respect to the device contact geometry. Since any vertical misalignment would likely be the same for all junctions this rules out any possible influence of any such misalignment. Thus we have clearly shown the intrinsic magneto-chiral origin of the JDE effect. In addition, the absence of the magneto-chirality of MoTe₂ and WTe₂-absent control devices allows one to further exclude the possibility of the non-zero vertical (misalignment) field. We have added this discussion in the revised manuscript (Supplementary Section S9).

4. What is the temperature for the data shown in Fig. 1c? The transition from the superconductive to the normal state is rather rounded, which might indicate that these curves were not taken at base temperature. Maybe the junction is overdamped because of the high temperature? In brief, the authors should show IV-curves at base temperature.

The I - V curves presented in Fig. 1c were measured at the base temperature (i.e. 20 mK) and they were well fitted by a standard formula for overdamped junctions, thereby allowing us to define I_c of the junction accurately.

5. The order of the presentation in the paper could be improved. Fig. 2 gives and discussed values for θ_{MC} , but only in Fig. 3 it is explained how these are obtained.

We appreciate the reviewer's suggestion. Please note that regarding θ_{MC} , we define θ_{MC} and describe our experimental method for obtaining θ_{MC} of our JJs when we introduce Fig. 1 (page 3). Thus, we prefer to keep the current order of the presentation flow as this reflects the order in which we carried out our experiments.

6. Fig. 2 compares junctions of different barrier thickness. However, besides barrier thickness, also temperature is changed. It would be better to show a consistent dataset in which only one parameter at a time is changed.

At the base temperature (of our measurement setup), a heating effect is unavoidable if the applied dc current to JJs goes beyond a few hundred μ A. To avoid this unintentional heating effect, we had no choice but to measure 2 nm and 7 nm thick barrier JJs at 200 mK.

Nonetheless, the I - V curves of 7 nm thick barrier JJs reveal no significant difference between 200 and 20 mK (See Fig. R3), as would be expected for the measurement temperature of $\leq 0.3T_c$. This explains why the critical currents of 2 nm (a) and 7 nm (b) thick barrier JJs measured at 200 mT are still much higher than those of 60 nm (c) thick barrier junction at 20 mK. We have included these data in our revised supplemental data (Supplementary Information S7). We thank the reviewer for encouraging us to include these data.

Fig. R3. Current-Voltage I - V curves of a van der Waals JJ with 7 nm thick T_d - WTe_2 barrier measured at $T = 20$ (black) , 200 mK (green) with (a) $\mu_0 H_{\parallel} = 0$ mT and (b) $\mu_0 H_{\parallel} = 40$ mT.

7. Fig. 4, give the values of T_c .

We have added the T_c values in Fig. 4 of our revised manuscript (Page 16).

8. It is surprising that the possibly most interesting dataset (on the twisted double-barrier) is hidden in the Extended Data. Why? In any case, the discussion of this sample should be extended, to provide sufficient information. Referring to point 2, what is the angular dependence of the SDE efficiency expected for the twisted bilayer case? Which function was used to fit the data in Extended Fig. 2c, and what is the physical model behind the fit function? It seems to be not a single sine function, and appears to be not even periodic with 360deg? A more detailed discussion is required.

We thank the referee for this excellent suggestion which we are happy to follow. We have included these data in the main text in a new figure 5 and 6. In a twisted double-barrier junction, *the vector addition of the internal crystal fields of the top and bottom WTe₂ layers*, respectively, with a twist angle between two layers leads to a new artificial polar axis of the entire Josephson barrier. In this sense, one can expect that the θ_{MC} is determined with respect to this new artificial polar axis. As shown in the Extended data Fig. 2c, diode efficiency of a twisted WTe₂ double-barrier JJ shows sinusoidal behaviour and diode efficiency of the JJ goes to zero when the magnetic field is applied to the parallel direction of the vector addition of the internal crystal field of top and bottom WTe₂ layers. This means that the magneto-chirality of twisted-double barrier JJ can be determined by the artificial polar axis of the barrier.

9. Besides, when was the α of the two barrier layers measured? Before or after the assembly of the single layers into the stack?

We first identified the crystal orientation of two chosen WTe₂ flakes through their elongated shapes which tends to be along the a -axis [Choi et al. *Nat. Mater.* **19**, 974-979 (2020)], and then carefully twisted and stacked one on top of the other to realize confirmed $\theta_{twist} \approx 90^\circ$. After completing all the transport measurements necessary, we finally confirmed the respective crystal orientations of the top and bottom WTe₂ flakes by a means of angle-resolved polarized Raman spectroscopy.

10. SI chapter S6: if there is no barrier, why does the device behave as if there was a barrier? What is acting as a barrier here? Is the NbSe₂ oxidized?

Due to the existence of a vdW gap between adjacent NbSe₂ electrodes, the WTe₂-absent NbSe₂/NbSe₂ devices can even behave as Josephson junctions (JJs). There is no reason to invoke oxidation. Indeed one can find a previous/similar study on the vdW NbSe₂/NbSe₂ JJs [Yabuki et al. *Nat. Commun.* **7** 10616 (2016)]. We have now included a reference to this paper in our revised manuscript.

In addition to reviewer's comment, we found some typos and have corrected in the revised manuscript.

Yours sincerely

Reviewers' Comments:

Reviewer #1:

Remarks to the Author:

I acknowledge the rebuttal letter of the authors which answers to most of the points risen by the referees but I cannot be satisfied by the new version of the paper which is only slightly modified compared to the previous one. In particular authors claim to have provided better averaged data in this new version compared to the previous one. However there is absolutely no difference between the data shown in this new version of the paper and the previous one! The quantity measured is the switching current which needs to be properly averaged in order to obtain the critical current of the junction. This work apparently was not done which explains the poor quality of the data presented. Regarding the discussion on SINIS-like junctions, I am also not convinced by the discussion of the authors who mention a "SINIS" contribution whereas barriers at NS interfaces are known to strongly decrease the amplitude of the Josephson current

They should eventually write instead:

"The presence of vdW gaps at the interfaces between the WTe₂ flake and the two adjacent NbSe₂ electrodes is expected to give rise to tunnel barriers at the NS interfaces. Therefore the junctions have to be considered as SINIS junctions which is corroborated by the exponential decay of the RNI_c product as a function of the thickness of the WTe₂ flake shown in the SM"

The SM section 5 should be corrected as well. The characteristic decay length of the exponential is indeed not simply the superconducting coherence length ξ_S but also depends of the transmission of the barriers. Even more important, such an exponential behaviour is indeed not expected in the absence of tunnel barriers at the NS interface for ballistic junctions where instead the product $RNI_c = \pi\Delta$ (Δ is the superconducting gap of the S electrodes) independent of the junction length for short junctions $L < \xi_S$ and decreases as $1/L$ in the long junction limit. This behavior is quite different from what is observed by the authors.

In conclusion the paper cannot be published in the present form. The minimum requirement is a correct discussion of the SINIS behavior of the junctions.

Reviewer #2:

Remarks to the Author:

I find the author's responses to the reviewers to be acceptable and recommend the manuscript for publication. I will ask that the authors include the temperature at which the IV's were taken in Figure 1c in the Figure panel itself or at least in the caption to make the information as accessible as possible to the reader.

Reviewer #3:

Remarks to the Author:

I have carefully read the rebuttal letter of the authors, and while I think that they have replied well to some of the raised issues, I also have to say that in my opinion their reply to some other points is not sufficient. I would therefore recommend a revision of the manuscript. In detail,

Ad 1), the authors write:

"To reflect this comment, we have expanded the introduction section of our manuscript by over-viewing the relevant recent studies."

Honestly, I could not find such an expanded review of the literature in the revised version of the manuscript. The authors should collocate their work within the literature, give credit to relevant previous work, and underline the novelty of their own work. The revised introduction of the manuscript does not sufficiently address these points.

Ad 2)

I am not sure that I entirely agree with the arguments of the authors, or maybe I'm just missing the point. The authors claim that Baumgartner et al. discuss interface inversion symmetry breaking. However, Baumgartner et al. discuss the impact of spin-orbit coupling on the magneto-chiral anisotropy, and thus an intrinsic property of the semiconductor. Indeed, the section headline reads "Impact of lattice orientation on magnetochiral anisotropy". I would like to add that I understand that the present manuscript goes beyond the results by Baumgartner et al., and the data presented in Figure 3 probably justify publication in Nature Communications, but still, I would ask the authors (a) to give credit to Baumgartner et al. by citing and discussing their work, and (b) to make an effort to underline the novelty of the present results in comparison. Besides, there is a paper by Kouwenhoven's group (arXiv 2211.14283) which comes to similar conclusions as Baumgartner et al. and which could be cited, as well.

Ad 8)

While the discussion in the rebuttal letter is sufficient to understand how the authors analyzed the data on the twisted double-barrier junction, none of this has been included in the manuscript. I strongly recommend doing this so that the readers can follow the argument. A closer look at Fig. 6(c) reveals that the fit is not purely sinusoidal, but there is a constant offset, as well. The maximum value of the curve is +4.5%, while the minimum value is about -5.2%. Thus, there is a constant contribution to η which is independent of θ . How do the authors explain this?

Ad 10)

The paper by Yabuki et al. properly explains why the authors could create a junction without a WTe₂ barrier: they have either misaligned or misoriented the two flakes or increased the van der Waals gap between them. I would suggest including this information in the SI. Contrary to what the authors state, I could not find the reference to Yabuki et al. in the manuscript or the SI.

Detailed response to referee's comments.

In the following, we provide detailed point-by-point responses to the referees' comments (referees' comments in blue; our responses in black).

A. Response to the First Referee:

I acknowledge the rebuttal letter of the authors which answers to most of the points risen by the referees but I cannot be satisfied by the new version of the paper which is only slightly modified compared to the previous one. In particular authors claim to have provided better averaged data in this new version compared to the previous one. However there is absolutely no difference between the data shown in this new version of the paper and the previous one! The quantity measured is the switching current which needs to be properly averaged in order to obtain the critical current of the junction. This work apparently was not done which explains the poor quality of the data presented.

We thank the reviewer for his/her rigorous review. To satisfy the referee's expectation, we have performed additional measurements on a *newly* fabricated MoTe₂ JJ (Fig. R1a) rather than presenting more averaged data of the existing MoTe₂ junction. As can be seen in Fig. R1, the obtained Fraunhofer interference patterns from this newly fabricated junction are well defined (Fig. R1b) and importantly, the almost absence of magneto-linearity (Fig. R1c) and magneto-chirality (Fig. R1d) is clearly probed. We believe that these results provide a better quality of the data and strongly support our claim, that is, the intrinsic Josephson diode effect derived from the Josephson barrier's crystal structure. The new results have been updated in our revised manuscript (Supplementary Information S4).

Fig. R1. a, Optical micrograph of $1T'$ -MoTe₂ Josephson junctions. Scale bar is 5 μm . Note that unlike the orthorhombic phase T_d -WTe₂, the monoclinic phase $1T'$ -MoTe₂ is centrosymmetric. **b**, Positive and negative Josephson critical current I_c^+ (green) and $|I_c^-|$ (orange) *versus* in-plane magnetic field $\mu_0 H_{\parallel}$ for a 6 nm thick $1T'$ -MoTe₂ barrier in a NbSe₂/ $1T'$ -MoTe₂/NbSe₂ Josephson junction. The measurement was conducted at $T = 2$ K. Josephson diode efficiency ($\eta = \frac{I_c^+ - |I_c^-|}{I_c^+ + |I_c^-|}$) as a function of magnetic field strength (**c**) and angle (**d**) for the NbSe₂/ $1T'$ -MoTe₂/NbSe₂ Josephson junction.

Regarding the discussion on SINIS-like junctions, I am also not convinced by the discussion of the authors who mention a ‘‘SINIS’’ contribution whereas barriers at NS interfaces are known to strongly decrease the amplitude of the Josephson current

They should eventually write instead:

‘‘The presence of vdW gaps at the interfaces between the WTe₂ flake and the two adjacent NbSe₂ electrodes is expected to give rise to tunnel barriers at the NS interfaces. Therefore the junctions have to be considered as SINIS junctions which is corroborated by the exponential decay of the $R_N I_c$ product as a function of the thickness of the WTe₂ flake shown in the SM’’

The SM section 5 should be corrected as well. The characteristic decay length of the exponential is indeed not simply the superconducting coherence length ξ_S but also depends of the transmission of the barriers. Even more important, such an exponential behaviour is indeed not expected in the absence of tunnel barriers at the NS interface for ballistic junctions where instead the product $R_N I_c = \pi \Delta$ (Δ is the superconducting gap of the S electrodes) independent of the junction length for short junctions $L < \xi_S$ and decreases as $1/L$ in the long junction limit. This behavior is quite different from what is observed by the authors.

We appreciate the referee for giving us an opportunity to go through how vdW gaps contribute to the transport phenomena of our vertical NbSe₂/WTe₂/NbSe₂ JJs. We do agree on the referee’s point that the vdW gap formed at either side of the WTe₂ Josephson barrier is responsible for the exponential decay of characteristic voltage of WTe₂ JJs as a function of the WTe₂ barrier thickness and thereby, our NbSe₂/WTe₂/NbSe₂ JJs need to be considered as ‘*SINIS junctions*.’ We have further realized from a previous study on lateral Nb/WTe₂/Nb JJs [*Natl. Sci. Rev.* **7**, 1468 (2020)] that in the case of ballistic junctions with transparent/direct contacts, the barrier-thickness-dependent characteristic voltage is expected to follow the $1/L$ dependence in the long junction limit.

As suggested by the referee, we have corrected the discussion of the role of vdW gaps in both the main text and Supplementary Information S5 of our revised manuscript.

In conclusion the paper cannot be published in the present form. The minimum requirement is a correct discussion of the SINIS behavior of the junctions.

We thank again the reviewer for his/her rigorous review of our manuscript. We hope the revised manuscript by adding new results and a more likely explanation will satisfy the referee’s expectation and finally recommend our manuscript for publication in Nature Communications.

B. Response to the Second Referee:

I find the author's responses to the reviewers to be acceptable and recommend the manuscript for publication. I will ask that the authors include the temperature at which the IV's were taken in Figure 1c in the Figure panel itself or at least in the caption to make the information as accessible as possible to the reader.

We appreciate the referee for his/her recommendation of our manuscript for publication in Nature Communications. To reflect the reviewer's comment, we have added the measurement temperature for I - V curves in the Fig.1c.

C. Response to the third Referee:

I have carefully read the rebuttal letter of the authors, and while I think that they have replied well to some of the raised issues, I also have to say that in my opinion their reply to some other points is not sufficient. I would therefore recommend a revision of the manuscript. In detail,

We thank the reviewer for his/her positive evaluation of our manuscript and constructive feedback.

1. the authors write:

“To reflect this comment, we have expanded the introduction section of our manuscript by over-viewing the relevant recent studies.”

Honestly, I could not find such an expanded review of the literature in the revised version of the manuscript. The authors should collocate their work within the literature, give credit to relevant previous work, and underline the novelty of their own work. The revised introduction of the manuscript does not sufficiently address these points.

2. I am not sure that I entirely agree with the arguments of the authors, or maybe I'm just missing the point. The authors claim that Baumgartner et al. discuss interface inversion symmetry breaking. However, Baumgartner et al. discuss the impact of spin-orbit coupling on the magneto-chiral anisotropy, and thus an intrinsic property of the semiconductor. Indeed, the section headline reads “Impact of lattice orientation on magnetochiral anisotropy”. I would like to add that I understand that the present manuscript goes beyond the results by Baumgartner et al., and the data presented in Figure 3 probably justify publication in Nature Communications, but still, I would ask the authors (a) to give credit to Baumgartner et al. by citing and discussing their work, and (b) to make an effort to underline the novelty of the present results in comparison. Besides, there is a paper by Kouwenhoven's group (arXiv 2211.14283) which comes to similar conclusions as Baumgartner et al. and which could be cited, as well.

We appreciate the reviewer's suggestions and valuable comments to further improve our paper. To reflect the above comments 1 and 2, we have expanded and rewritten the introduction section of our manuscript as follows.

As pointed out by the reviewer, Baumgartner *et al.* already first showed a possible correlation between the bulk inversion symmetry breaking (*i.e.* Dresselhaus-type) and the supercurrent non-reciprocity. In the case of interface inversion symmetry breaking (*i.e.* Rashba-type), the associated spin-orbit (SO) field in k -space is fundamentally isotropic. When the Josephson barrier possesses the

Dresselhaus-type symmetry breaking in addition to the Rashba-type symmetry breaking, the overall SO field becomes anisotropic, making the magneto-chiral anisotropy of supercurrent non-reciprocity dependent of the Josephson barrier's crystal structure. However, the strength of Dresselhaus-type SO field in Baumgartner *et al.* was found to be quite small ($< 10\%$ at most) as compared with that of Rashba SO field. To the best of our knowledge, supercurrent non-reciprocity derived *solely* from the bulk inversion symmetry breaking of the Josephson barrier is yet to be demonstrated. In our paper, we demonstrate that the supercurrent non-reciprocity along the vertical direction is intimately connected and is highly dependent on the crystal structure of the T_d -WTe₂ barrier, providing the first experimental realization of the intrinsic JDE that results from the *pure* bulk inversion symmetry breaking of the T_d -WTe₂ barrier.

3. While the discussion in the rebuttal letter is sufficient to understand how the authors analyzed the data on the twisted double-barrier junction, none of this has been included in the manuscript. I strongly recommend doing this so that the readers can follow the argument.
A closer look at Fig. 6(c) reveals that the fit is not purely sinusoidal, but there is a constant offset, as well. The maximum value of the curve is +4.5%, while the minimum value is about -5.2%. Thus, there is a constant contribution to η which is independent of θ . How do the authors explain this? What is the influence of a possible small vertical component of the applied in-plane magnetic field due to a misalignment? Could this explain some of the findings?

As suggested by the referee, we have added our discussion of how we have analysed data on the twisted double-barrier junction in our revised manuscript (Page 7).

As for the constant offset in Fig. 6(c), most of our junctions studied showed non-zero constant offset that is irrelevant to the intrinsic JDE of magneto-chiral origin [*Nat. Mater.* **21**, 1008 (2022)]. Such a constant offset originates highly likely from the asymmetric interface of the junctions as discussed in Wu *et al.* [*Nature* **604**, 653 (2022)]. Furthermore, a small OOP component of the applied magnetic field due to a misalignment can, in principle, generate the IP-field-angle-independent offset. We have added this discussion in our revised manuscript (Page 5).

4. The paper by Yabuki *et al.* properly explains why the authors could create a junction without a WTe₂ barrier: they have either misaligned or misoriented the two flakes or increased the van der Waals gap between them. I would suggest including this information in the SI. Contrary to what the authors state, I could not find the reference to Yabuki *et al.* in the manuscript or the SI.

We appreciate the reviewer for his/her careful reading. We have added the reference to Yabuki *et al.* in Supplementary Information of our revised manuscript.

Yours sincerely

Jae-Keun Kim & Stuart S. P. Parkin

Reviewers' Comments:

Reviewer #1:

Remarks to the Author:

In this new version authors provide data on a new MoTe₂ sample. However this data raises new questions which should be addressed:

-This data is much less noisy than previous one and does not seem at first glance to exhibit any reciprocal asymmetry. However it should be plotted on a smaller Y scale in order to compare with data measured on WTe₂ samples and I am not sure it will be different from the data shown on the 60nm thick WTe₂ sample with a value of asymmetry reaching a few % at 5 mT.

-Moreover I recently realised that the 1T'-MoTe₂ phase is not stable at low temperature for thin flakes see PHYSICAL REVIEW B 97, 041410(R) (2018). Are the authors sure that they could really prepare a centrosymmetric 1T'-MoTe₂ phase?

Otherwise I agree with the other modifications on the paper specially the discussion concerning the existence of barriers at the NbSe₂WTe₂ interface. I am however not sure that the statement . "Given the distinctively different barrier-thickness dependence of ϕ_0 on whether the junction is in the ballistic or diffusive regime, such a deviation is likely related to a ballistic-to-diffusive (long-junction) transition²⁴⁻²⁶." still holds in the presence of these barriers since samples clearly do not behave at standard SNS junctions with low contact resistances.

Finally I also noticed an erroring Fig3c on the thickness of the sample 60 instead of 6nm.

Reviewer #3:

Remarks to the Author:

I have carefully read the rebuttal letter of the authors, and I think that this time they have replied well to all the issues raised by this referee. I have also checked that all changes were implemented in the manuscript. Thus, I now do recommend acceptance of the manuscript.

Detailed response to referee's comments.

In the following, we provide detailed point-by-point responses to the referees' comments (referees' comments in blue; our responses in black).

A. Response to the First Referee:

In this new version authors provide data on a new MoTe₂ sample. However this data raises new questions which should be addressed:

-This data is much less noisy than previous one and does not seem at first glance to exhibit any reciprocal asymmetry. However it should be plotted on a smaller Y scale in order to compare with data measured on WTe₂ samples and I am not sure it will be different from the data shown on the 60nm thick WTe₂ sample with a value of asymmetry reaching a few % at 5 mT.

We thank the reviewer for his/her rigorous review. For fair comparison, we believe that the 11 nm thick MoTe₂ junction needs to be compared with the 7 nm thick WTe₂ junction as they have comparable Josephson critical supercurrent densities. Note that as the Josephson barrier thickness [Supplementary Information Section S5] increases the Josephson critical supercurrents exponentially decay.

-Moreover I recently realised that the 1T'-MoTe₂ phase is not stable at low temperature for thin flakes see PHYSICAL REVIEW B 97, 041410(R) (2018). Are the authors sure that they could really prepare a centrosymmetric 1T'-MoTe₂ phase?

We appreciate the referee for pointing out a possible crystal phase transition in MoTe₂ flakes. We would first like to note that a recent Raman spectroscopy on the thickness evolution of the phase transition [*Phys. Rev. B* **102**, 54103 (2020)] showed a significant suppression of the phase transition for thin flakes. In particular, below the thickness of 20 nm, all the investigated thin flakes revealed signatures of the 1T' phase even at low temperatures. These results suggests that the phase transition is mostly allowed for thick bulk flakes and is suppressed for thinner flakes of MoTe₂. A quite recent Raman study [*ACS Nano* **15**, 2962 (2021)] also pointed out the greatly suppressed phase transition between 1T' and T_d phases in MoTe₂ thin flakes at low temperatures.

Nevertheless, to reflect the referee's comment, we have rewritten Supplementary Information S4 of our revised manuscript as follows.

S4. Control vdW Josephson junction with a **MoTe₂ barrier**

Fig. S4. Suppression of magneto-linearity and magneto-chirality in supercurrents through a **locally centrosymmetric** vdW Josephson barrier **a**, Optical micrograph of a MoTe₂ Josephson junctions. Scale bar is 5 μm. Note that unlike the orthorhombic phase T_d -WTe₂^{S1,S2,S3}, the monoclinic phase $1T'$ -MoTe₂ is centrosymmetric^{S4}. Our MoTe₂ flake (with around 11 nm) may possess a *mixed phase* of T_d and $1T'$ at low temperatures as recently discussed in detail in^{S5,S6}. **b**, Positive and negative Josephson critical current I_c^+ (green) and $|I_c^-|$ (orange) versus in-plane magnetic field $\mu_0 H_{||}$ for a 11 nm thick MoTe₂ barrier in a NbSe₂/MoTe₂/NbSe₂ Josephson junction. The measurement was conducted at $T = 2$ K. Josephson diode efficiency ($\eta = \frac{I_c^+ - |I_c^-|}{I_c^+ + |I_c^-|}$) as a function of magnetic field strength (c) and angle (d) for the NbSe₂/ MoTe₂/NbSe₂ Josephson junction.

Otherwise I agree with the other modifications on the paper specially the discussion concerning the existence of barriers at the NbSe₂/WTe₂ interface. I am however not sure that the statement . “Given

the distinctively different barrier-thickness dependence of ϕ_0 on whether the junction is in the ballistic or diffusive regime, such a deviation is likely related to a ballistic-to-diffusive (long-junction) transition²⁴⁻²⁶” still holds in the presence of these barriers since samples clearly do not behave at standard SNS junctions with low contact resistances.

We appreciate the referee for this comment. We have added a brief discussion of another possible origin of the non-monotonic thickness-dependent diode efficiency in our revised manuscript as follows (page 7).

As the c -axis mean free path is close to the c -axis coherence length of our WTe₂ JJs, one should also consider a short-to-long junction transition³² *within* the ballistic limit (Supplementary Information S5). To clarify whether the non-monotonic thickness-dependent η^* results from the ballistic-to-diffusive transition²⁴⁻²⁶ or the short-to-long ballistic transition³², a further study is necessary in the future.

Finally I also noticed an erroring Fig3c on the thickness of the sample 60 instead of 6nm.

We appreciate the reviewer for his/her careful reading. We have corrected this typo in our revised manuscript.

B. Response to the third Referee:

I have carefully read the rebuttal letter of the authors, and I think that this time they have replied well to all the issues raised by this referee. I have also checked that all changes were implemented in the manuscript. Thus, I now do recommend acceptance of the manuscript.

We appreciate the referee for his/her recommendation of our manuscript for publication in Nature Communications.

Yours sincerely

Jae-Keun Kim & Stuart S. P. Parkin

REVIEWERS' COMMENTS

Reviewer #1 (Remarks to the Author):

Dear Editor

The authors answered to my last remarks and corrected the manuscript accordingly.

I consider that the manuscript now deserves publication in Nature communications.

Best regards

Hélène Bouchiat

Detailed response to referee's comments.

In the following, we provide detailed point-by-point responses to the referees' comments (referees' comments in blue; our responses in black).

A. Response to the First Referee:

The authors answered to my last remarks and corrected the manuscript accordingly. I consider that the manuscript now deserves publication in Nature communications.

We appreciate the referee for his/her recommendation of our manuscript for publication in Nature Communications.

Yours sincerely

Jae-Keun Kim & Stuart S. P. Parkin